# On the Interplay Between Roughness and Elastic Modulus at the Nanoscale: A Methodology Study with Bone as Model Material

**DOI:** 10.3390/jfb16080276

**Published:** 2025-07-29

**Authors:** Alessandro Gambardella, Gregorio Marchiori, Melania Maglio, Marco Boi, Matteo Montesissa, Jessika Bertacchini, Stefano Biressi, Nicola Baldini, Gianluca Giavaresi, Marco Bontempi

**Affiliations:** 1Scienze e Tecnologie Chirurgiche, IRCCS Istituto Ortopedico Rizzoli, 40136 Bologna, Italy; alessandro.gambardella@ior.it (A.G.); melania.maglio@ior.it (M.M.); gianluca.giavaresi@ior.it (G.G.); marco.bontempi@ior.it (M.B.); 2Scienze e Tecnologie Biomediche e Nanobiotecnologie, IRCCS Istituto Ortopedico Rizzoli, 40136 Bologna, Italy; marco.boi@ior.it (M.B.); matteo.montesissa@ior.it (M.M.); nicola.baldini@ior.it (N.B.); 3Department of Surgery, Medicine, Dentistry and Morphological Sciences with Interest in Transplant, Oncology and Regenerative Medicine, University of Modena and Reggio Emilia, 41125 Modena, Italy; jessika.bertacchini@unimore.it; 4Department of Cellular, Computational and Integrative Biology, University of Trento, 38123 Trento, Italy; stefano.biressi@unitn.it; 5Department of Biomedical and Neuromotor Sciences, University of Bologna, 40100 Bologna, Italy

**Keywords:** nanoindentation, roughness, cortical bone, tissue biomechanics, atomic force microscopy, image analysis

## Abstract

Atomic force microscopy (AFM)-based nanoindentation enables investigation of the mechanical response of biological materials at a subcellular scale. However, quantitative estimates of mechanical parameters such as the elastic modulus (E) remain unreliable because the influence of sample roughness on E measurements at the nanoscale is still poorly understood. This study re-examines the interpretation of roughness from a more rigorous perspective and validates an experimental methodology to extract roughness at each nanoindentation site—i.e., the local roughness γ_s_—with which the corresponding E value can be accurately correlated. Cortical regions of a murine tibia cross-section, characterized by complex nanoscale morphology, were selected as a testbed. Eighty non-overlapping nanoindentations were performed using two different AFM tips, maintaining a maximum penetration depth of 10 nm for each measurement. Our results show a slight decreasing trend of E versus γ_s_ (Spearman’s rank correlation coefficient ρ = −0.27187). A total of 90% of the E values are reliable when γ_s_ < 10 nm (coefficient of determination R^2^ > 0.90), although low γ_s_ values are associated with significant dispersion around E (γ_s_ = 0) = E_0_ = 1.18 GPa, with variations exceeding 50%. These findings are consistent with a qualitative tip-to-sample contact model that accounts for the pronounced roughness heterogeneity typical of bone topography at the nanoscale.

## 1. Introduction

In the field of biological materials, modern approaches to the study of the local mechanical properties are increasingly focused on reducing both the mechanical stresses applied to elicit a response from the tissue and the spatial scale of investigation. The latter involves both the penetration depth of the probe within the bone matrix and the area of tissue probed. Quantitative mechanical results obtained from this volume can thus be considered representative of specific compositional and structural characteristics of the biomaterial at the micro- or nanoscale. Revealing mechanical properties at the nanoscale is important (i) for the early diagnosis of pathological and degenerative mechanisms (i.e., first or indirect, subtle, alterations of the extra-cellular matrix structure and composition), e.g., testing small tissue biopsies/explants from in vivo (animal or clinical) studies [1,2,3]; and (ii) for the surface characterization of scaffolds and substrates during in vitro studies for tissue engineering/regenerative medicine, e.g., in mechanotransduction [4,5,6,7]. Atomic force microscopy (AFM)-based nanoindentation represents a unique and well-established tool capable of providing spatial maps of the mechanical characteristics of surfaces at loading forces in the nanonewton range, and of simultaneously investigating the surrounding topographic features with nanometric spatial resolution [8]. However, in the case of shallow nanoindentations, mechanical data obtained by AFM have a generally poor quantitative value, because of the limited understanding of how the sample’s topography affects the mechanical response of the surface. For this reason, flat surfaces, i.e., surfaces with roughness much smaller than the penetration depth, are recommended to extract reliable nanoindentation data [9].

Unfortunately, this recommendation is often impossible to follow with shallow nanoindentations. This constitutes a gap in the knowledge of surface layer mechanics, in that most of the surfaces of biomaterials exhibit important roughness at the nanoscale, and this roughness cannot be decoupled from the measure of local mechanical properties. A further complication arises from the fact that roughness is generally a scale-dependent quantity; on average, it tends to an asymptotic value at large scales, while decreasing as a power law towards small scales. In spite of this, several nanoindentation works have provided surface “roughness” as referring to the value calculated as an average extracted from topographic images of an arbitrary size; however, it is clear that with shallow nanoindentations, position (of the nanoindentation) and scale (contact area between the indenter and the surface) should be considered in roughness calculation. In other words, the local roughness of each nanoindentation area must be considered. Extraction of such local roughness, with the aim to ensure reliable quantification of mechanical properties, is the object of the present study, which also provides insights into the interrelation between local roughness and elastic modulus in the experimental context of AFM-based nanoindentations.

Among biological materials, the multiscale hierarchical structure of bone—which is crucial to its biological and mechanical properties—results in a complex, heterogenous morphology at the micro- and nanoscale with strong variations in roughness. Probing the mechanical properties of bone at nanometric depths has great potential for exploring tissue-scale phenomena such as the mechanics of the cell traction forces and response to mechanical stimuli [10,11,12,13,14], as well as response of the tissue to micro- and nano-wear [15,16,17,18,19]. In this respect, shallow nanoindentation of bone implies dependence of the roughness on the position (i.e., on the osseous district mechanically tested), making it an ideal test bed for a correlation study between mechanical properties and local roughness.

In this work, we employ a section of mouse cortical bone previously processed for hard-resin embedding. It is underlined that the choice to use a processed section of bone rather than a fresh sample is primarily due to the need for a surface with an aspect ratio suitable for AFM measurements, however, without loss of generality as concerns the search for an interrelation between roughness and elastic modulus. Specifically, we perform measurements at eighty non-overlapping positions within a region of interest (ROI) measuring 25 × 25 µm^2^, selected through Fourier-Transform Infrared (FTIR) analysis to exclude surface contributions from impurities. Shallow indentations can benefit from using nanometrically sharp indenters, as they enhance spatial resolution in topographic mode and improve force sensitivity in nanoindentation mode. Accordingly, this study employs nanoindentation with two nanometrically sharp tips, evaluating the elastic modulus at a fixed penetration depth of 10 nm, under the assumption of elastic deformation in an infinite, isotropic half-space. It is important to note that the widely used Hertz model for elastic deformation assumes that deformations are infinitesimally small relative to the size of the indenter. Therefore, a more appropriate model—capable of accounting for cases where the indenter size and penetration depth are comparable—will be used in this work.

The objectives of this study are to quantitatively investigate the following:


(i)The impact of local roughness on the reliability of elastic modulus measurements, where reliability is assessed using the coefficient of determination (R^2^) from the fit of the experimental loading force curve at locations with measured roughness.(ii)The strength of the correlation between elastic modulus and local roughness, evaluated by non-parametric statistical analysis.


Extracting roughness from a nanometric area presents several experimental challenges, including precise positioning, spatial resolution, sampling limitations, and thermal drift mitigation. These issues will be addressed in dedicated sections.

## 2. Materials and Methods

### 2.1. Histological Processing for Polymethyl Methacrylate Embedding

A tibia explanted from a 90-day-old C57BL/6J wild-type mouse (Charles River Laboratories, Wilmington, MA, US, no. 000664) was used. The mouse had access to food and water ad libitum and was kept at a constant temperature on a 12:12 h light/dark cycle. The animal was sacrificed by competent personnel through euthanasia by inhalation of CO_2_, followed by cervical dislocation, in accordance with current regulations. After harvesting, the bone segment was fixed in neutral buffered formalin for a few hours and stored in 70% ethanol. The sample was processed for hard resin (polymethyl methacrylate, PMMA) embedding [20].

Briefly, samples were dehydrated in increasing concentrations of ethanol (two steps in 96% ethanol, followed by two steps in absolute ethanol), each step lasting one day. Samples were then infiltrated with methyl methacrylate-based solutions until complete polymerization of the resin was achieved (Merck, Shuchardt, Hohenbrunn, Germany). Using a microtome with rotative diamond blade, Leica SP1600 (Leica Microsystems Spa, Milan, Italy), the sample was cut along transversal planes at the level of metaphyseal and diaphyseal areas, obtaining sections representative of cortical bone tissue. A section was glued to plastic microscope slides using a cyanoacrylate-based glue, ground up to a thickness of (50 ± 10) μm and smoothed by a polishing system, Saphir 550 (Direct Industry, Marseille, France), using velvety cloths to reduce possible cutting artifacts. An unstained section was then used for FTIR and AFM analyses.

### 2.2. FTIR Setup

Infiltration of PMMA into bone osteons may occur and can be influenced by tissue porosity; therefore, FTIR spectroscopy was employed to detect its presence. The bone sections were scanned through Spectrum 2 coupled with a microscope, a Spotlight 200i instrument, in Attenuated Total Reflectance mode (ATR-FTIR) (PerkinElmer, Waltham, MA, USA). Images with different zooms were acquired by camera view of a microscope (field of view of the camera 450 × 360 μm) through an image survey as a mosaic of adjacent camera view images, which give an image of a larger area (Figure 1a,b). The following scanning parameters were used: acquisition range 4000–700 cm^−1^, aperture size 25 μm × 25 μm, resolution 4 cm^−1^, 32 scans, and data interval 1 cm^−1^. Reference PMMA spectra were previously collected in a zone made of pure polymer, i.e., where bone was not present.

### 2.3. Targeting the ROI: FTIR Preliminary Analysis

Prior to AFM characterization, the composition of bone tissue was assessed via FTIR analysis to confirm the absence of PMMA. A 25 µm × 25 µm large ROI was selected and analyzed at five locations (see Figure 1) to investigate the possible presence of PMMA. The different spectra colors correspond to the marker where the acquisition was performed, as displayed by the image. The bone tissue was composed by the organic (mainly collagen) and inorganic components (carbonated hydroxyapatite, HA) as expected. In fact, bands at 1650 (peptide C=O stretching vibration of the collagen, Amide I), 1550 (C–N stretching and N–H in-plane bending modes, Amide II), 1452 (asymmetrical stretching mode of CO_3_ν_3_), 1415 (symmetrical stretching mode of CO_3_ν_3_), 1240 (PO_2_ antisymmetric stretching of Amide III), 1015 (antisymmetric stretch mode of ν_3_PO_4_), 965 (symmetric stretching of ν_1_PO_4_) and 870 (CO_3_ν_2_) cm^−1^ were noticed [21]. In general, from the PMMA spectra (the black curve in Figure 1c), it can be observed that the main PMMA bands (around 1724 and 1145 cm^−1^) did not overlap with those of bone tissue, which allows for the precise identification of the two different materials.

However, small overlaps in the region connected with the carbonate (1500 ÷ 1400 cm^−1^) and phosphate (1100 ÷ 1000 cm^−1^) groups of bone tissue carbonated HA can be noticed. In fact, bands around 1724 and 1145 cm^−1^ were linked to the C=O of ester and C-O-C stretching vibration, respectively [22]. The absence of these two bands in the spectra confirmed the lack of PMMA in the ROI analyzed.

### 2.4. AFM Setup Calibration and Operation

AFM measurements were performed by a SMENA system (NT-MDT, Moscow, Russia) equipped with an upright optical microscope. The basic principle of AFM-based nanoindentation is as follows: a tip (i.e., a spherical tip of radius r) is brought into contact with the sample surface, and the deflection of the cantilever, which is a lever of stiffness k, is measured as a function of the distance, generating a force–distance indentation curve. This curve is then converted into a force–penetration depth (F-h) curve by a calibration operation [23,24]. The elastic modulus at the indentation point E can be then extracted from the F–h curve using appropriate fitting models (Figure 2). Two spherical NSG30 tips (NT-MDT, Moscow, Russia) with a resonant frequency in the range of 140–390 kHz and a nominal curvature radius of r = 10 nm were used, named tip 1 and tip 2 in the following. The cantilevers’ stiffnesses were k_1_ = 11.96 N/m and k_2_ = 13.38 N/m, respectively. *k* was measured via software (NOVA, MT-MDT, Moscow, Russia) according to Sader [24]. The cantilevers’ deflection sensitivity was calibrated according to a previous work of this group [23]. The tip integrity was checked via z-axis calibration on a TGS1 calibration grating (NT-MDT, Moscow, Russia; grid TGZ1 with height (21 ± 1) nm). All topographic images containing nanoindentation points were recorded at a 1024-pixel resolution and acquired in tapping mode of operation. For all the indents, the maximum loading force F_max_ was triggered to about 220 nN to obtain loading curves like in Figure 2, reaching a penetration depth of h_max_ = (10.0 ± 0.5) nm. A triangular load–unload profile was used, with loading and unloading times set to 1 s each. The F–h curves were processed with an elastic (Hertzian) model built for a spherical probe of radius *r* and h ≤ 4*r* [25]. The Poisson’s ratio of bone, which is used in the calculation of E, was assumed to be 0.3, according to previous works [23,26].

### 2.5. AFM Data Processing

Within the ROI established by the FTIR analysis, all the AFM data (topographic images and curves) were processed using Python (version 3.10) to calculate, respectively, local roughness and E. As sketched in Figure 3a, the script receives as an input the set of indentation force–distance curves and the acquired topography. The indentation curves also contain the coordinates of their acquisition positions. In this way, it is possible to know where the indentations on the topography were taken.

At each indentation coordinate, which is the center of the maximum indentation area A_c_ (with r = 10 nm, A_c_ ~ 300 nm^2^), a square sub-region of interest A_γ_ was extracted from the image. Then, the (local) k-th roughness value γ_sk_ was calculated within A_γ_ according to the expression:(1)$$\gamma_{s_{k}}=\sqrt{\frac{1}{n_{rows}n_{cols}}\sum_{i=1}^{n_{rows}}\sum_{j=1}^{n_{cols}}h_{ijk}^{2}}$$
where *i* and *j* are row and column indices, *n_rows_* and *n_cols_* are the number of rows and columns in A_γ_, respectively. The parameter *h_ijk_*, expressed in nm, is the k-th height at coordinates *i,j* in the topographic image.

Thermal drift during scan was assumed according to Sadeghzadeh [27], i.e., assuming a 2° temperature change during a 10 min scan of the sample, the corresponding maximum thermal drift error is ~3.3 nm. Thus, the process of image recording and curve acquisition on the image was regulated on this basis. A rectangular 5 µm × 0.5 µm image at a 1024-pixel resolution was acquired. Setting the scan speed at 1 Hz, this required 1.75 min. Then, on this image, the acquisition of an array of 10 spectra started a few seconds later. It is noticed that, as the loading–unloading operation takes 1 s for each curve, most of the time is taken by the control electronics for positioning the tip during each indent, requiring ~1 min to complete spectra acquisition. In this condition, the maximum thermal drift error should be less than 1 or 2 nm. Arguments based on the uncertainty of tip positioning and thermal deflection of the cantilever suggest that this value could be underestimated by a factor [28,29]; thus, in this work, A_γ_ ~ 10A_c_ was chosen, i.e., A_γ_ ~2900 nm^2^ (11 pixel × 11 pixel). This also allows us to ensure reliable sampling for γ_s_ calculation by Equation (1).

### 2.6. Software and Statistics

Before Python elaboration, the AFM images were processed (1st order filtering) by Gwyddyon software (version 2.58, Prague, Czech Republic). A total of 40 + 40 non-overlapped indents were operated for each tip to reduce consumption phenomena as possible, and to have a statistically relevant amount of data points. Force curves were exported as raw data from the acquisition software NOVA (NT-MDT, Moscow, Russia) and processed by Python (version 3). Data representation and Shapiro–Wilk tests at a 0.05 level (*p*-value, *p*) were carried out by Origin (version 7.5). Spearman’s rho (ρ) correlations and *p*-value (*p*), Mann–Whitney tests (*p*-value, *p*), and a clustering operation (k-means function) were performed by Matlab software (2024a, The MathWorks, Natick, MA, USA).

## 3. Results

Shapiro–Wilk tests suggest that the E and γ_s_ datasets were not normally distributed, with the exception of the E dataset for tip 1 (tip 1: *p* = 0.05553 and 0.00013 for E and γ_s_, respectively; tip 2: *p* = 0.02 and 0.00005 for E and γ_s_, respectively). First, we tested the influence of tip consumption during nanoindentations on the measure of the E. Therefore, the latter is plotted against the indentation number (1 ≤ N ≤ 40) for each dataset obtained from tip 1 and tip 2 (Figure 4a,b), together with the corresponding linear fittings.

The slight descending (ascending) monotonic trends in the E vs. N plots are not significant (Spearman’s coefficients, ρ = 0.01295 and 0.30525, respectively, for tip 1 and tip 2), suggesting the absence of biases affecting the measure of E. Nevertheless, it is interesting to compare these graphs with the corresponding γ_s_ vs. N plots (Figure 4c,d) and their trends. γ_s_ did not change significantly during the measurement (ρ = 0.29231 and −0.26623, respectively, for tip 1 and tip 2). However, the slight ascending (descending) trends in the γ_s_ vs. N plots appear associated with the corresponding, opposite trends in the E vs. N plots.

A statistical correlation between morphological variation and mechanical measurement is observed when the γ_s_ and E distributions are compared, as in Figure 5, where the results of non-parametric significance tests are reported. One notices a remarkable difference (about one order of magnitude in roughness) between the morphologies encountered by tip 1 and tip 2.

Such a difference is highly significant (*p* < 0.000001, Figure 5a) and corresponds to a significant difference (*p* = 0.02656) between the corresponding E distributions. To investigate whether the spherical Hertzian fitting was influenced by γ_s_, we plotted the coefficients of determination of the spherical fittings (R^2^) against γ_s_ (Figure 6).

It was found that 69% of cases with R^2^ > 0.95 occurred when γ_s_ ≤ 5 nm. Generally speaking, this result agrees with the universally accepted indication that flat surfaces are favorable to reliable measurements of E; however, 97% of the E values were estimated with R^2^ > 0.90, suggesting that the spherical model is a good fitting of the experimental curves in the measured range of γ_s_ values. It should be noted that the uncertainties on the tips’ radii imply that the potential influence of the tip shape cannot be quantified. However, the results obtained so far suggest that—in the present experiment—γ_s_ essentially dominates upon tip geometry in determining the E values. Under the assumption that γ_s_ is the only variable influencing the measurement of E, one can consider the union of the (E, γ_s_) pairs obtained from tip 1 and tip 2, and plot them in Figure 7a. Note that the complete E dataset is normally distributed at the 0.05 level (*p* = 0.2659), while the homologous γ_s_ dataset is not (*p* < 0.00001). Thus, Spearman’s correlation was used. Note that the apparent lack of normality in the γ_s_ distribution was likely due to the strongly heterogeneous distribution of the asperities, by which the two tips encountered regions differing by about one order of magnitude in roughness. As already suggested by the analysis in Figure 4 and Figure 5, a weakly decreasing trend between E and γ_s_ can be observed (ρ = −0.27187, *p* = 0.01471). If the y-intercept of the linear fit (E(0) = E_0_ ~ 1.18 GPa) is taken as representative of a flat surface, we see that when γ_s_ ~ h_max_, a diminution of E by about 30% with respect to E_0_ can be observed.

In Figure 7b, data clustering is reported. Clustering operation is aimed at revealing patterns and trends in the E vs. γ_s_ datasets. Specifically, the operation partitioned data into two mutually exclusive clusters in which objects within each cluster were as close to each other as possible, and as far from objects in the other cluster as possible. Each cluster is characterized by its centroid or center point.

In this respect, by looking at Figure 7b, two considerations arise: (i) γ_s_ is the parameter that mainly defines potential cluster regions, which suggests that only E values extracted from loci with γ_s_ ≤ 5 nm can be considered reliable. Moreover, most of the data are concentrated in the “quasi-flat” region, γ_s_ ≤ 2 nm, showing the highest scattering of the corresponding E values. (ii) Cluster centroids highlight a descending (E vs. γ_s_) trend, in line with the monotonic decrease highlighted in Figure 7a.

## 4. Discussion

### 4.1. Comparison with Literature on Shallow Nanoindentation of Bone

Several studies employing AFM-based indentation have underscored the importance of minimizing the influence of surface morphology when assessing mechanical properties. For example, Wang et al. emphasized the need for using flat surfaces when comparing the mechanical properties of different scaffolds for bone-marrow-derived human mesenchymal stem cells [30]. Similarly, Asgari et al. investigated the mechanical properties of bone tissue in adult mouse femurs using AFM nanoindentation [26]. In this work, F_max_ = 350 nN for cortical bone and 150 nN for trabecular bone were applied, with h < 50 nm to avoid nonlinear and inelastic responses at higher strain levels. E in the range of 0.1–1 GPa was reported in trabecular and cortical bones, respectively, which is consistent with our findings. In this study, bone sections were polished prior to nanoindentation testing to minimize residual surface roughness.

Similarly, Xu et al. examined the mechanical properties of various polymeric substrates for rat bone-marrow-derived mesenchymal stem cells. Although their study revealed differences in both mechanical response and surface roughness among the polymers, no explicit correlation between the two parameters was reported [31]. To our knowledge, a review of the literature on bone nanoindentation suggests that, in studies focusing on shallow indentation depths, the relationship between surface roughness and mechanical properties has not been thoroughly investigated.

For instance, Katsamenis et al. examined the mechanical and failure behavior of both tough and weak human cortical bone at the osteonal, micro-, and tissue levels using AFM-based nanoindentation [32]. In their study, F_max_ = 100 nN and 0 < h < 10 nm. Nanoindentation was employed to assess mechanical variations occurring between lamellar and interlamellar regions. These regions exhibited distinct mechanical properties as well as differing features in AFM topography. The authors attributed these differences to a combination of architectural and compositional factors.

Grue et al. developed a method for the facile production of mineralized collagen fibril sheets with tunable mechanical properties for bone scaffold applications. The nanomechanical properties of individual collagen fibrils extracted from these scaffolds were evaluated using AFM cantilevers with a spring constant of 7.2 N/m. Their study employed PeakForce Quantitative Nanomechanical Mapping mode to accurately determine the tip radius, which was found to be (10.06 ± 1.80) nm at a 10 nm indentation depth and (17.57 ± 3.35) nm at 20 nm—conditions closely matching those of our experiments, where F_max_ ~ 300 nN. Here, E was calculated using an adhesion-based contact model, yielding values in the range of 0.1–1 GPa. According to the authors, the adhesion-based approach may help minimize the influence of roughness, which was estimated to fall in the nanometer range [33].

Hongo et al. investigated osteocytes and their lacunae in the femora and tibiae of both wild-type and pathological mice following administration of human parathyroid hormone (PTH). AFM nanoindentation was employed to assess the mechanical properties with spatial maps of moduli of the osteocytic–perilacunar matrix in PTH-treated samples. While some correlation between topographic features and mechanical properties was visually evident, no explicit analysis or quantification of roughness or its potential impact on nanoindentation results was provided [34].

Cisneros et al. focused on evaluating the mechanical properties of osteonal cortical bone from bovine tibia at the lamellar level [35]. In this study, bone specimens were first polished and immersed in saline to achieve a smooth surface, then dried and analyzed via AFM for both topographic imaging and localized nanoindentation of individual lamellae. AFM probes with a nominal tip radius of 150 nm and spring constant 80 N/m were used. To the best of our knowledge, this is the only study to quantitatively emphasize the role of surface roughness in relation to indentation contact depth and its impact on nanoscale mechanical measurements. Specifically, the authors recommended maintaining indentation depths between 10 and 50 nm to avoid inconsistent results arising from localized peak-to-valley interference.

### 4.2. General Considerations, Limitations, and Benefits of the Study

The influence of surface roughness on the measurement of mechanical properties is well established in traditional nanoindentation, both for spherical [36] and non-spherical indenters [37]. In general, increased surface roughness leads to greater measurement uncertainty and a reduction in the measured elastic modulus—results that agree very well with those reported in this study. Therefore, one may conclude that the scale reduction investigated here acts primarily as a geometric factor affecting the measurement of the elastic modulus. With this in mind—under the hypothesis of homogeneous and isotropic surface—the simplest, qualitative graphical representation of the actual contact area between the indenter and the surface can be provided in the cases of γ_s_ ≪ h (Figure 8a) and γ_s_ ~ h (Figure 8b). Such a representation effectively highlights how the presence of void regions beneath the spherical indenter results, for a given h, in a reduced surface contribution to E, as was observed experimentally. However, the uncertainty of the indenter’s shape and the absence of strong statistical correlations between E and γ_s_ observed in this study suggest that such a representation is somewhat simplistic, and thus, the measurement of mechanical properties at the nanoscale is influenced by multiple factors. In particular, shallow nanoindentations may amplify the effects of bone’s multiscale heterogeneity and anisotropy, leading to increased data variability even at a low γ_s_, as shown in Figure 7. This observation points to a limitation in the present study—and, more broadly, in studies that rely solely on surface roughness as the primary parameter for assessing mechanical properties at the nanoscale.

Compositional variations—specifically the mechanically distinct contributions of stiffer mineral and softer collagen components in bone—could not be disentangled from topographical nanoscale features [38], as also observed in a recent work by this group [23]. Moreover, spatial inhomogeneity and preferential orientation, inherent to bone’s hierarchical structure, are likely to influence the mechanical response [39,40].

To address these limitations, future developments of this research will integrate the automated image and force curve analysis workflow established in this study with finite element simulations. This combined approach will help to evaluate how mechanical properties vary under increasingly complex conditions of local roughness and structural heterogeneity.

The procedure developed here has the potential of investigating (bone) tissue on micrometric regions of interest, that is, to reveal the actual mechanical behavior of the extra-cellular matrix and not the apparent mechanical response of a macroscopic region (e.g., of spongy bone where macro-architecture, i.e., pores and trabecular arrangements, has a strong influence). Since an important part of this study consists in the implementation of software for the processing of topographic images, the methodology can be easily extended to entire topographic regions of the sample, e.g., through an immediate comparison between the γ_s_ map and the corresponding E map, allowing us to aggregate the mechanical data of various microscopic regions of interest until reaching the macroscopic, but still intrinsic, material properties. It is the case of averaging various cortical bone zones vs. trabecular zones, or of aggregating and comparing zones of different bone areas (e.g., metaphysis vs epiphysis), that is the ongoing work of this group for demonstrating potential applications, also with considering the use of micro-axial tomography for a complementary evaluation of the bone mineral density within the ROI [41]. The possibility to reveal early (nanoscopic and microscopic) and advanced (macroscopic) degenerations (e.g., osteoporosis, aging, genetic musculoskeletal disease, fracture risk) on small tissue biopsies/explants from in vivo (animal or clinical) studies highlights the methodology potential towards clinical applications that will be developed in the future.

## 5. Conclusions

The relationship between the elastic modulus and local roughness was investigated during shallow indentations of cortical murine bone using atomic-force-microscopy-based nanoindentation. The findings support the widely accepted principle that flatter surfaces—those with roughness much smaller than the indentation depth—yield more reliable mechanical measurements. However, in shallow nanoindentations, the effects of multiscale heterogeneity and anisotropy may be amplified compared to traditional nanoindentation, leading to greater variability in the measured elastic modulus even at low roughness, and reducing the statistical correlation between modulus and roughness. Non-uniform crystallinity and structural anisotropy, which cannot be disentangled from topography, likely contribute to this effect.

Conversely, a more complex topography highlights a geometric effect: when roughness becomes comparable to the indentation depth, the actual contact area is significantly smaller than the theoretical contact area. This discrepancy reduces the contribution of the surface to the measured mechanical response, leading to an underestimation of the elastic modulus. Our findings quantitatively showcase the utility of automated procedures to improve the accuracy of nanoindentation data analysis, and provide guidance for extracting more reliable mechanical data under unconventional operating conditions on materials with complex topographies at the nanoscale, of which bone is a relevant example.

## Figures and Tables

**Figure 1 jfb-16-00276-f001:**
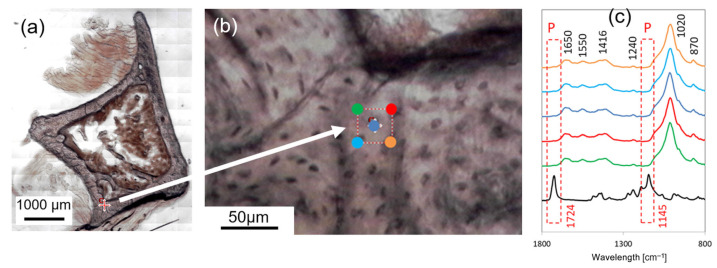
(**a**) Optical image of the slice, collected as a mosaic of adjacent camera view. The red cross identifies the ROI analyzed. (**b**) Zoomed-in image of the ROI, together with the positions where five FTIR spectra were collected. (**c**) The five spectra (colors corresponding to the previous image) with indication of the bands relevant to the analysis. The PMMA spectrum (black) is also reported for comparison.

**Figure 2 jfb-16-00276-f002:**
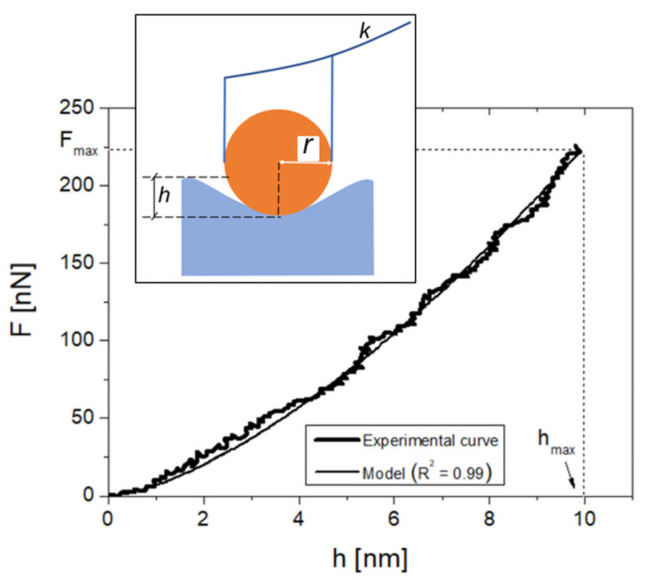
A representative experimental F–h curve obtained in this work, with indication of F_max_ and h_max_, together with the related curve fitting. In the inset is a sketch of the nanoindentation process by a cantilever-tip system of stiffness *k* and radius *r* that penetrates to a depth of h nanometers below the surface.

**Figure 3 jfb-16-00276-f003:**
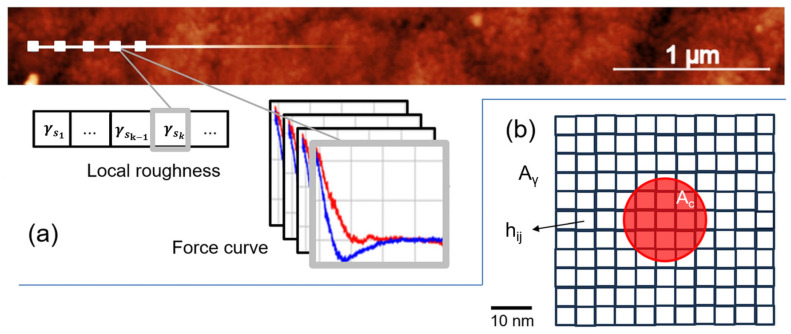
(**a**) Rectangular topographic image with indication (white squares) of the array of nanoindentation positions where local roughness γ_s_ is extracted at the k-th position (γ_sk_) within the array of force curves. In the sketch, the red (blue) is the loading (unloading) nanoindentation curve. (**b**) Zoomed view of a white square (**a**) on a proportioned scale showing the maximum contact area A_c_ and the sampled area A_γ_, where γ_s_ was calculated, with indication of the generic h_ij_ height.

**Figure 4 jfb-16-00276-f004:**
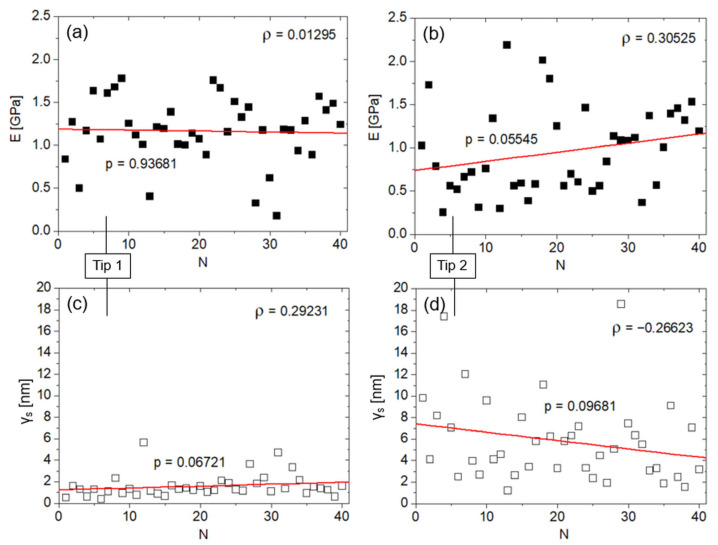
(**a**) Plot of γ_s_ vs. N and (**b**) plot of E vs. N for tip 1 (black squares). Red lines indicate monotonic trends, while graphs report the corresponding *p*-values and Spearman’s ρ correlation values. (**c**) Plot of γ_s_ vs. N and (**d**) plot of *E* vs. N for tip 2 (white squares) with corresponding monotonic trends (red lines), *p*- and Spearman’s ρ correlation values.

**Figure 5 jfb-16-00276-f005:**
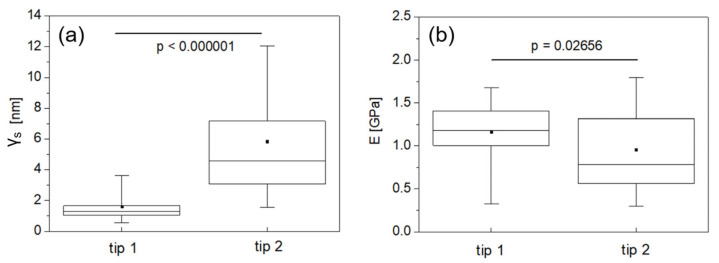
(**a**) Box plot of (**a**) γ_s_ data and (**b**) E data shown separately for tip 1 and tip 2, reporting non-parametric significance test results.

**Figure 6 jfb-16-00276-f006:**
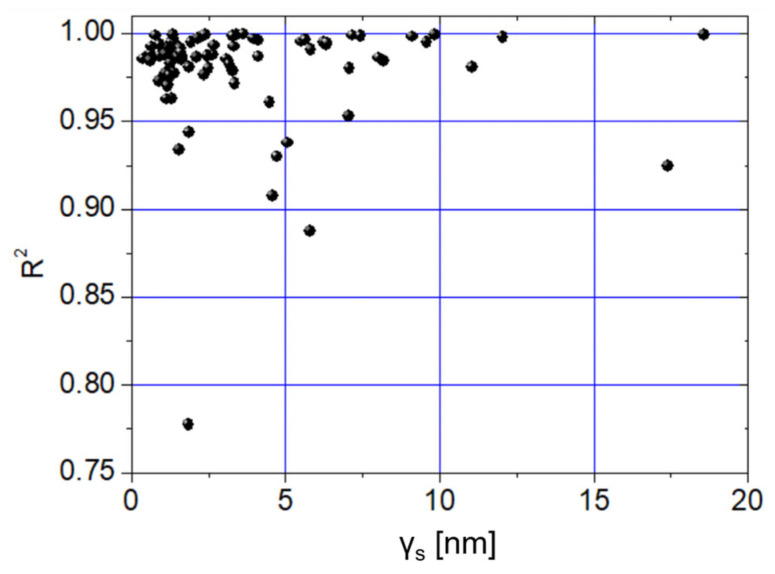
Plot of R^2^ against γ_s_ for the eighty curves collected by nanoindentations operated by tip 1 and tip 2.

**Figure 7 jfb-16-00276-f007:**
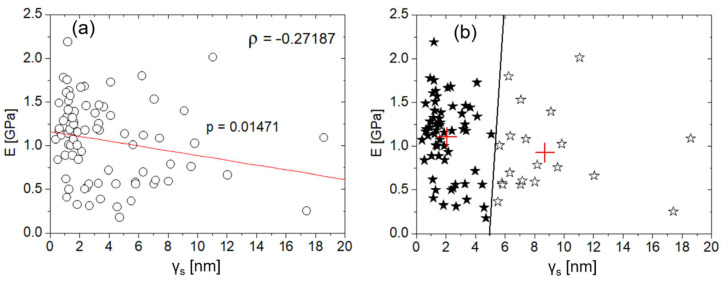
(**a**) Plot of E against γ_s_ (white circles). The red line indicates the monotonic trend. Spearman’s *p*- and ρ -values are also reported. (**b**) The same data as (**a**), with indication of the centroids (red crosses) extracted by clustering analysis. The two clusters around the respective centroids are indicated by black and white stars.

**Figure 8 jfb-16-00276-f008:**
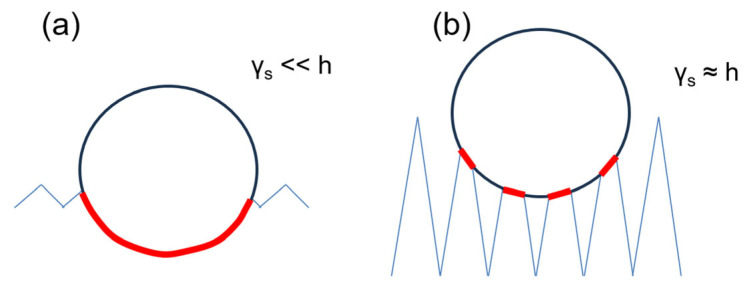
Actual contact region (in red) between the indenter and the surface in the cases (**a**) γ_s_ ≪ h and (**b**) γ_s_ ~ h.

## Data Availability

The original contributions presented in the study are included in the article, further inquiries can be directed to the corresponding author.

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
