# Peer review of "On the Interplay Between Roughness and Elastic Modulus at the Nanoscale: A Methodology Study with Bone as Model Material"

_jfb, 2025, doi:10.3390/jfb16080276_

Round 1
Reviewer 1 Report
Comments and Suggestions for Authors
This study aimed to investigate the correlation between nanoscale surface roughness and elastic modulus in mouse cortical bone using atomic force microscopy (AFM)-based nanoindentation. Although this research holds substantial merit, there are some methodological and interpretational concerns require further attention and clarification.
- To improve the document's overall clarity and English language proficiency, a comprehensive edit would be helpful.
- The abstract should including key methodological details such as sample size, and preparation protocols. These additions would help readers better evaluate and build upon the study's findings
- The introduction should clearly articulate the existing gaps in knowledge that this study addresses. Adding a concise paragraph highlighting the novel aspects of this work would help readers better appreciate the study's significance and originality.
- Authors should explain whether the measurements were obtained from a single tibia or multiple specimens. Reporting the sample size (number of bones tested) in the abstract aand methodology section should be included, this would help assess potential biological variability and strengthen the generalizability of the findings.
- Line 114: The authors define dehydration using increasing ethanol concentrations followed by methacrylate-based infiltration. For reproducibility, precise ethanol gradients and methacrylate solution concentrations should be detailed.
- The methodology section should include justification for the sample size used in the study. A brief explanation of statistical considerations or power calculations would help readers evaluate the robustness of the findings
- The discussion could better articulate how these nanoscale findings might translate to macroscopic bone properties or clinical conditions (e.g., osteoporosis, aging).
- To strengthen the literature context, consider citing recent studies (2020–2025)

The English could be improved to convey the research more clearly.
Author Response
Letter of Response to the Reviewers
We would like to thank all the Reviewers for their careful revision of our manuscript. In the following, a point-to-point reply to all of their concerns/observations/suggestions is provided. We do hope the changes made have fulfilled their requests, highlighted the novelty and clarity of our work and made the manuscript suitable for publication in Journal of Functional Biomaterials. All the relevant changes made have been marked in yellow in this new version of the manuscript.
Reviewer 1
This study aimed to investigate the correlation between nanoscale surface roughness and elastic modulus in mouse cortical bone using atomic force microscopy (AFM)-based nanoindentation. Although this research holds substantial merit, there are some methodological and interpretational concerns require further attention and clarification.
We thank the Reviewer for recognizing the merit of our study. In the following, we provide a point-to-point reply to all the raised concerns. Nevertheless, in this new version of the manuscript we have much better underlined that our aim was not to obtain generalized results on bone, but rather to test a novel methodology of characterization of biomaterials specifically developed for shallow, or surface-layer nanoindentations. In this context, bone—a material with highly complex topological features at the micro- and nanoscale- was an ideal test bed for our methodology, which allows to compare nanoscale roughness and the local elastic modulus -measured at the same point- with an unprecedented degree of accuracy. We acknowledge that this aspect was not sufficiently emphasized in the previous version, which may have led to legitimate expectations regarding the statistical robustness of the measurements. Thus, in this new version we have better clarified the main objectives of our work; in particular, the importance of decouple roughness and mechanical properties during nanoindentation of biomaterials has now been much better emphasized. To do this, we have first modified the title of the manuscript making it more suitable to emphasize what the reader should expect from the presented work. The Reviewer will found detailed all the relevant modifications made according to Her/His observations.
- To improve the document's overall clarity and English language proficiency, a comprehensive edit would be helpful.
The language has now been carefully checked and improved throughout the manuscript.
- The abstract should including key methodological details such as sample size, and preparation protocols. These additions would help readers better evaluate and build upon the study's findings
The abstract has now been modified according to the concerns of all the Reviewers. To emphasize the concept of local roughness, which is a key concept of our study, we found appropriate to adopt a new symbol to distinguish it from roughness calculated on an arbitrary region. Yet, the role of bone as test bed for our measurements has been concisely underlined and other changes made to improve the clarity and completeness of information, and the help the reader to grasp the take-home message of the article.
- The introduction should clearly articulate the existing gaps in knowledge that this study addresses. Adding a concise paragraph highlighting the novel aspects of this work would help readers better appreciate the study's significance and originality.
We thank the Reviewer for this suggestion. We have modified the Introduction to better underline gaps in the existing knowledge, and to highlight the originality of our work. (Lines 64-78).
- Authors should explain whether the measurements were obtained from a single tibia or multiple specimens. Reporting the sample size (number of bones tested) in the abstract and methodology section should be included, this would help assess potential biological variability and strengthen the generalizability of the findings.
The methodology proposed in the manuscript was validated on the cortical region of a single tibia, where a PMMA-free region (ROI) was identified by FTIR, and then measured by AFM nanoindentation. As now better stated, our aim was to validate a methodology, and not to obtain generalized results strictly related to bone, that would require comparison between different district osseous, and consideration of another important parameter characteristic of the nanoindentation volume, i.e., bone mineral density [please see our contribution to the European Society of Biomechanics July 2025 congress*]. Thus, this work represents a starting point for further and more extensive characterizations, including analysis of several samples and more district osseous.
*https://www.conftool.com/esb2025/index.php/ESB2025_D2_10_Marchiori.pdf?page=downloadPaper&filename=ESB2025_D2_10_Marchiori.pdf&form_id=840.
Line 114: The authors define dehydration using increasing ethanol concentrations followed by methacrylate-based infiltration. For reproducibility, precise ethanol gradients and methacrylate solution concentrations should be detailed.
The required details have now been concisely added to Section 2.1
- The methodology section should include justification for the sample size used in the study. A brief explanation of statistical considerations or power calculations would help readers evaluate the robustness of the findings
As specified, in this study the statistical sample size is given by the number of indentations on which the correlation between local roughness and elastic modulus is evaluated. The chosen number (N = 80) represents a compromise between the need to avoid tip consumption problems (wear), limiting to 40 indentations per tip, and the need to sample the roughness within the ROI at an acceptable level. A concise sentence with justification has now been added (Lines 259-261).
- The discussion could better articulate how these nanoscale findings might translate to macroscopic bone properties or clinical conditions (e.g., osteoporosis, aging).
We have now added a part on the possible clinical benefits with possible translation from micro-to macroscopic scale of our study (Lines 438-455) at the end of the Discussion section.
- To strengthen the literature context, consider citing recent studies (2020–2025)
More recent studies have been considered in this revised version, so that several references have been replaced with more recent ones.
Reviewer 2 Report
Comments and Suggestions for Authors
The article is interesting and contains relevant information, but it requires a few modifications:
-
It is recommended to structure the abstract using the following components: background (if necessary), objectives, methodology, results, and conclusion. In its current form, the abstract is somewhat confusing and overly focused on highlights.
-
For quantitative analyses, it is advisable to include numerical values in the results section of the abstract, in addition to the p-value.
-
Review the formatting throughout the manuscript, as there are inconsistencies in paragraph indentation—some paragraphs begin with spacing while others do not.
Author Response
Letter of Response to the Reviewers
We would like to thank all the Reviewers for their careful revision of our manuscript. In the following, a point-to-point reply to all of their concerns/observations/suggestions is provided. We do hope the changes made have fulfilled their requests, highlighted the novelty and clarity of our work and made the manuscript suitable for publication in Journal of Functional Biomaterials. All the relevant changes made have been marked in yellow in this new version of the manuscript.
Reviewer 2
We thank the Reviewer for appreciating our work and Her/His valuable suggestions. We have modified the main text accordingly.
The article is interesting and contains relevant information, but it requires a few modifications:
- It is recommended to structure the abstract using the following components: background (if necessary), objectives, methodology, results, and conclusion. In its current form, the abstract is somewhat confusing and overly focused on highlights.
The Abstract has now been modified according to the Reviewer’s suggestion.
- For quantitative analyses, it is advisable to include numerical values in the results section of the abstract, in addition to the p-value.
New quantitative insights have been concisely added in the Abstract.
- Review the formatting throughout the manuscript, as there are inconsistencies in paragraph indentation—some paragraphs begin with spacing while others do not.
The manuscript has now been reviewed and also the inconsistencies in paragraph indentation have been fixed.
Letter of Response to the Reviewers
We would like to thank all the Reviewers for their careful revision of our manuscript. In the following, a point-to-point reply to all of their concerns/observations/suggestions is provided. We do hope the changes made have fulfilled their requests, highlighted the novelty and clarity of our work and made the manuscript suitable for publication in Journal of Functional Biomaterials. All the relevant changes made have been marked in yellow in this new version of the manuscript.
Reviewer 2
We thank the Reviewer for appreciating our work and Her/His valuable suggestions. We have modified the main text accordingly.
The article is interesting and contains relevant information, but it requires a few modifications:
- It is recommended to structure the abstract using the following components: background (if necessary), objectives, methodology, results, and conclusion. In its current form, the abstract is somewhat confusing and overly focused on highlights.
The Abstract has now been modified according to the Reviewer’s suggestion.
- For quantitative analyses, it is advisable to include numerical values in the results section of the abstract, in addition to the p-value.
New quantitative insights have been concisely added in the Abstract.
- Review the formatting throughout the manuscript, as there are inconsistencies in paragraph indentation—some paragraphs begin with spacing while others do not.
The manuscript has now been reviewed and also the inconsistencies in paragraph indentation have been fixed.
Reviewer 3 Report
Comments and Suggestions for Authors
Thank you for the opportunity to review your work on the influence of surface roughness on the measured elastic modulus of bone at the nanoscale. Understanding the mechanisms behind adaptive phenomena through bone remodelling is a key issue in contemporary bone research. This study demonstrates the importance of detailed validation of measurement methods in this field. However, the paper could be strengthened by adding the following points to the discussion.
Introduction
P2L44-
The background of the study is described in detail, making it easy for the reader to understand. However, the authors should elaborate on the current problems and issues that need to be resolved in related research. In addition, the clinical benefits should be explained.
P3L95
The hypothesis of the problem the authors wish to elucidate should be stated.
P3L101-104
This sentence should be listed in the discussion section.
Methods
P3L109-
The protocol for this animal experiment is described, but the process by which the experimental animals were euthanized without stressing them is not described. The ethics approval number should also be included.
P4L152-153
The meaning of this sentence should be more clearly explained to the reader.
P6L242
How was the sample size for this study calculated? This should be stated in this section.
P7L247
Matlab product information (trade name, company name, city, country) should be included.
P7L248
Is it necessary to state the cutoff value for significant differences (P<0.05) with respect to this statistical study?
Result
P8L294-
A detailed explanation of the results is provided. Although the Spearman correlation is examined, the lack of normality of continuous variables in the outcome data should be explained.
Discussion
Creating an indenter with atomically sharp edges is difficult, and it can be inferred that the indenter is subject to severe wear during indentation tests and lacks reproducibility.
What insights does this study provide regarding that issue?
What do the authors consider regarding the slippage of the indenter in the early stages of indentation and the sudden increase in load at the dented site due to measurements made at sites with poor bone surface parallelism?
P14L357-
My question is, do bone microcracks result from this gap in mechanical properties between cortical and trabecular bone? Please discuss any findings from this experiment that could offer insight.
Author Response
Letter of Response to the Reviewers
We would like to thank all the Reviewers for their careful revision of our manuscript. In the following, a point-to-point reply to all of their concerns/observations/suggestions is provided. We do hope the changes made have fulfilled their requests, highlighted the novelty and clarity of our work and made the manuscript suitable for publication in Journal of Functional Biomaterials. All the relevant changes made have been marked in yellow in this new version of the manuscript.
Reviewer 3
Thank you for the opportunity to review your work on the influence of surface roughness on the measured elastic modulus of bone at the nanoscale. Understanding the mechanisms behind adaptive phenomena through bone remodelling is a key issue in contemporary bone research. This study demonstrates the importance of detailed validation of measurement methods in this field. However, the paper could be strengthened by adding the following points to the discussion.
We warmly thank the Reviewer for recognizing the relevance of our work. In the following, we provide a point-to-point reply to all of Her/His concerns/suggestions.
Introduction
P2L44-
The background of the study is described in detail, making it easy for the reader to understand. However, the authors should elaborate on the current problems and issues that need to be resolved in related research. In addition, the clinical benefits should be explained.
We have modified the Introduction section to address this comment (lines 64-78), better elaborating on the issues that this study attempts to address, particularly concerning the problem of correctly comparing roughness and elastic modulus at the nanoscale. A new part (Lines 438-455) has been added to the Discussion section to highlight possible clinical benefits of our study.
P3L95
The hypothesis of the problem the authors wish to elucidate should be stated.
This has been done in the Introduction al lines 74-78 and, contextually, objectives stated at lines 104-109.
P3L101-104
This sentence should be listed in the discussion section.
The sentence has been moved further (re-elaborated in the Discussion (Lines 442-447) and Conclusions (Lines 473-476) sections.
Methods
P3L109-
The protocol for this animal experiment is described, but the process by which the experimental animals were euthanized without stressing them is not described.
We thank the Reviewer for requiring this detail; a sentence has now been added at Line 122-124.
The ethics approval number should also be included.
For the approval number, please see the Institutional Review Board Statement at the end of the manuscript.
P4L152-153
The meaning of this sentence should be more clearly explained to the reader.
This sentence has better explained (Lines 166-169).
P6L242
How was the sample size for this study calculated? This should be stated in this section.
Two lines have been added on this (Lines 259-261). Please see also reply to Reviewer 1 at point 5.
P7L247
Matlab product information (trade name, company name, city, country) should be included.
The required information has now been added.
P7L248
Is it necessary to state the cutoff value for significant differences (P<0.05) with respect to this statistical study?
We thank the Reviewer for this comment; we have fixed a 0.05 significance for the Shapiro-Wilks test, according to the Origin’s standard to suggest a decision on normality of the analysed distributions and consequently, on using non-parametric data testing. For other comparisons, the 0.05 cutoff is helpful in many biological and biomechanical context; however, for the reader’s convenience, we have specified all the p- and rho- values.
Result
P8L294-
A detailed explanation of the results is provided. Although the Spearman correlation is examined, the lack of normality of continuous variables in the outcome data should be explained.
A tentative explanation for the lack of normality of roughness is that it is not distributed homogeneously upon the surface, i.e., a difference of about one order of magnitude in roughness was measured between two different groups of regions within the ROI. This fact is -more likely- indicative of the heterogeneity of bone at the nanoscale and has now better underlined in the text (Line 316-319). We also suggested that roughness influenced the measured modulus more than geometrical effects due to tip shape, or to its consumption during scan/indentation, so it is likely that the lack of normality in the E values might follow the lack of normality in the roughness values.
Discussion
Creating an indenter with atomically sharp edges is difficult, and it can be inferred that the indenter is subject to severe wear during indentation tests and lacks reproducibility.
What insights does this study provide regarding that issue?
The Reviewer is perfectly right on the issue of having a very sharp indenter. This implies that, ultimately, the real shape of the tip is not known. Surely, knowing that exact shape is not possible. However, some considerations can be made:
- Even in a “rough” scenario, like the one analysed in this work, the measured moduli agree with the values obtained by previous literature on similar samples. This can be indicative of the fact that the tip, whose radius enters as a negative power into modulus calculation, does not differ to much in shape from the expected one.
- Using a tip for only 40 nanoindentation, like we did, may reduce the possibility of wear (and consequent distortion of the results). We have monitored this effect in the graphs of Figure 4.
The insight that this study provided is that, most likely, in the context analysed roughness has more influence on the measured E value than the tip shape (or, than possible changes of the tip shape).
What do the authors consider regarding the slippage of the indenter in the early stages of indentation and the sudden increase in load at the dented site due to measurements made at sites with poor bone surface parallelism?
We thank the Reviewer for raising these issues. Regarding slipping, it is likely that we did not experience this effect due to the shallow nature of the nanoindentation. Penetration of a few nanometers should guarantee a mostly elastic deformation of the material, as confirmed by the aspect of the F-h curves obtained in our work, which did not show any pop-in phenomena.
For the sudden increase in load due to bone surface non-parallelism, we may say that, since the investigated area is very small, it is likely that local roughness still prevailed over a possible surface tilt, masking its effect. On the other hand, on a larger scale, in our unfiltered images, the ratio between vertical variation and image width is usually about 100 nm / 10,000 nm, so only a very slight tilt is present. In any case, the Reviewer is right in recommending that these small-scale effects be considered, although the shape of the force curves and the consistency of the measured values with those reported in similar works remain the standard for comparison.
P14L357-
My question is, do bone microcracks result from this gap in mechanical properties between cortical and trabecular bone? Please discuss any findings from this experiment that could offer insight.
While demonstrating our procedure as valid and effective, we are already moving for integrating it with composition and structure measures at the microscale, and for applying it, e.g in comparing bone cortical and trabecular zones [contribute to the European Society of Biomechanics July 2025 congress*]. Therefore, our experiment could offer insight into cortical-trabecular transitions zones; for relating this information to a major risk for microcracking, various zones (only cortical, only trabecular, cortical-trabecular) should be investigated and cracked in a controlled way. To do so, our experimental apparatus (AFM-based nanoindentation) could not reach the needed loads, even if equipped with bigger tips and stiffer cantilevers. Thank you for the suggestion, we need to think about it.
*https://www.conftool.com/esb2025/index.php/ESB2025_D2_10_Marchiori.pdf?page=downloadPaper&filename=ESB2025_D2_10_Marchiori.pdf&form_id=840.
Round 2
Reviewer 1 Report
Comments and Suggestions for Authors
I recommend accepting the paper since all concerns have been properly addressed.
Reviewer 3 Report
Comments and Suggestions for Authors
Congratulations! All modification confirmed.